# Morphodynamic Response to Low-Crested Detached Breakwaters on a Sea Breeze-Dominated Coast

**Alec Torres-Freyermuth** [1,2,*], **Gabriela Medellín** [1,2], **Ernesto Tonatiuh Mendoza** [1,2], **Elena Ojeda** [1,2] **and Paulo Salles** [1,2]

1   Laboratorio de Ingeniería y Procesos Costeros, Instituto de Ingeniería, Unidad Académica Sisal, Universidad Nacional Autónoma de México, Sisal 97356, Mexico; GMedellinM@iingen.unam.mx (G.M.); EMendozaP@iingen.unam.mx (E.T.M.); EOjedaC@iingen.unam.mx (E.O.); PSallesA@iingen.unam.mx (P.S.)
2   Laboratorio Nacional de Resiliencia Costera, Laboratorios Nacionales CONACTY, Sisal 97356, Mexico
*   Correspondence: ATorresF@iingen.unam.mx; Tel.: +52-988-931-1000

**Abstract:** Low-crested detached breakwaters (LCDBs) have been widely employed as a mitigation measure against beach erosion. However, only a few studies have assessed their performance in sea-breeze-dominated environments. This work investigates the beach morphodynamics behind LCDBs deployed on a micro-tidal sea-breeze-dominated beach. The study area, located in the northern Yucatán peninsula, is characterized by low-energy, high-angle waves, which drive a persistent (westward) alongshore sediment transport ($O(10^4)$ m$^3$/year). High-resolution real-time kinematics global positioning system (GPS) beach surveys were conducted over a one-year period (2017–2018) to investigate the performance of LCDBs at three sites. Moreover, unmanned aerial vehicle flights were employed to evaluate far-field shoreline stability. Field observations revealed a distinct behavior in the three study sites, dependent on the breakwaters' transmission characteristics, geometry, stability, and shoreline orientation. Impermeable LCDBs, made of sand-filled geosystems, induced significant beach accretion (erosion) in up-(down-)drift areas. On the other hand, permeable LCDBs, made of Reef Ball™ modules, induced moderate beach changes and small erosion in down-drift areas owing to higher transmission coefficients. Measurements of LCDBs' freeboard height show that sand-filled geosystems' breakwaters presented a significant loss of sand during the study period, which explains the unexpected beach morphodynamic response on the lee side of the structure. Observations suggest that the study area is highly sensitive to the presence of LCDBs with low transmissivity.

**Keywords:** beach morphodynamics; UAV flights; beach surveys; Reef Balls™; sand-filled geosystems

## 1. Introduction

Sandy beaches and their associated front dunes provide both natural coastal protection against storm events and a habitat for different marine and terrestrial species. Therefore, beach erosion due to either natural or anthropogenic processes may cause environmental and economic impacts in the coast. This is particularly relevant in low-lying coastal areas, which are prone to climate change impacts, such as sea level rise and increasing storm activity [1].

Despite the increasing popularity of soft engineering [2,3], mixed soft–hard systems [4], and eco-engineering [5] solutions, so-called low-crested detached breakwaters (LCDBs), such as submerged rubble mounds, are popular in many locations around the world (e.g., [6]). The main parameters controlling shoreline response in the presence of LCDBs are distance offshore, length and orientation of the structure, transmission characteristics of the structure, depth at the structure, freeboard height, and wave characteristics [7,8]. Previous works have extensively investigated the stability, performance, and ecological impact of rubber-mound (conventional) LCDBs [8–11], but less

effort has been devoted to understanding the performance of unconventional LCDBs (e.g., sand-filled geosystems and artificial reefs) until more recently (e.g., [12–15]).

Shoreline response to LCDBs has been investigated by means of numerical models (e.g., [16]), physical model tests (e.g., [11]), and field observations (e.g., [9]). However, monitoring studies involving measurements of subsequent detailed bathymetry surveys and far-field effects are scarce, owing to both economic and technical constraints [9]. Predicting the morphodynamic response on the lee side of LCDBs is challenging [7]. The wave transmission, controlled by the freeboard level, permeability, and wave conditions, determines the size and location of the shoreline salient [16], and hence, a transmission coefficient must be incorporated in the design formulas [7]. Black and Andrews [17] investigated the salient amplitudes for islands and reefs separately and developed a power curve relationship similar to the one for emergent breakwaters presented by [18].

In recent years, the construction of breakwaters with the use of synthetic materials, such as geotextile tubes filled with sand, has become widespread in some countries owing to their lower permanent impact on natural coastal processes [15]. Other approaches have considered the use of modular structures, such as artificial reefs (e.g., Reef Ball™), that mimic the wave dissipation effects of natural coral reefs. Artificial reefs have become popular in tropical regions, because they provide other ecosystem services, such as a habitat for marine species, while enhancing shoreline stability [19]. The use of sand-filled geosystems increased in popularity in the state of Yucatán (México) in the early 2000s, partially because of governmental environmental agencies that consider this alternative to be a soft engineering solution [20].

This work aims to identify the role of changing wave conditions, LCDB type, geometry, freeboard elevation, and distance and orientation of the structure on the morphological response on the northern Yucatan coast by means of a monitoring program conducted from March 2017 to May 2018 at three different sites. The outline of this paper is the following. First, the study area and LCDB characteristics are described in Section 2. The beach-monitoring program conducted at the study sites is described in Section 3. The results of the beach morphodynamics are presented in Section 4. A discussion on the factors affecting the shoreline salient prediction in the study sites is presented in Section 5. Finally, concluding remarks are given in Section 6.

## 2. Study Area

### 2.1. Description

The study area is characterized by a mild-slope continental shelf and a micro-tidal regime [21]. Intense sea breeze events generate short-period, high-incidence angle NE waves that are present all year. Thus, sea breezes play an important role in nearshore hydrodynamics [22] and sediment transport [23]. Furthermore, Central American cold surge (CACS) events, associated with cold-front passages, generate NNW swell waves that occur during winter months [24]. Appendini et al. [25] found that the net potential (westward) of sediment transport in the area is approximately 35,000 m$^3$/year and that such value is highly sensitive to shoreline orientation.

The northern Yucatán peninsula has been experiencing beach erosion over the past few decades, mainly associated with coastal development [26]. The construction of eleven shelter ports and a major offshore port from the 1960s to the 1980s solved many socioeconomic problems, but, at the same time, motivated the deployment of hundreds of unauthorized groins by beach homeowners at locations affected by the impoundment of the ports' jetties [27]. The presence of coastal structures increased during the following decades until the authorities enforced such structure removal, mainly consisting of permeable groins, in the early 2000s. Beach nourishment combined with the groin removal provided satisfactory results along a 5-km stretch of coast [28]. However, coastal erosion has been exacerbated in certain areas due to the lack of a long-term beach nourishment program, the removal of the primary dune due to the growth of the coastal urban area, and the continuous deployment of unauthorized structures [29]. Meyer-Arendt [27] estimated erosion rates to be between 0.3 and 1.0 m/year west

of Progreso. A government-funded project considered the deployment of two breakwaters, made of sand-filled geosystems, as mitigation measures against beach erosion in two critical erosion spots located east of Progreso. At the same time, a privately funded project deployed breakwaters made of Reef Balls™ in the same area (Figure 1).

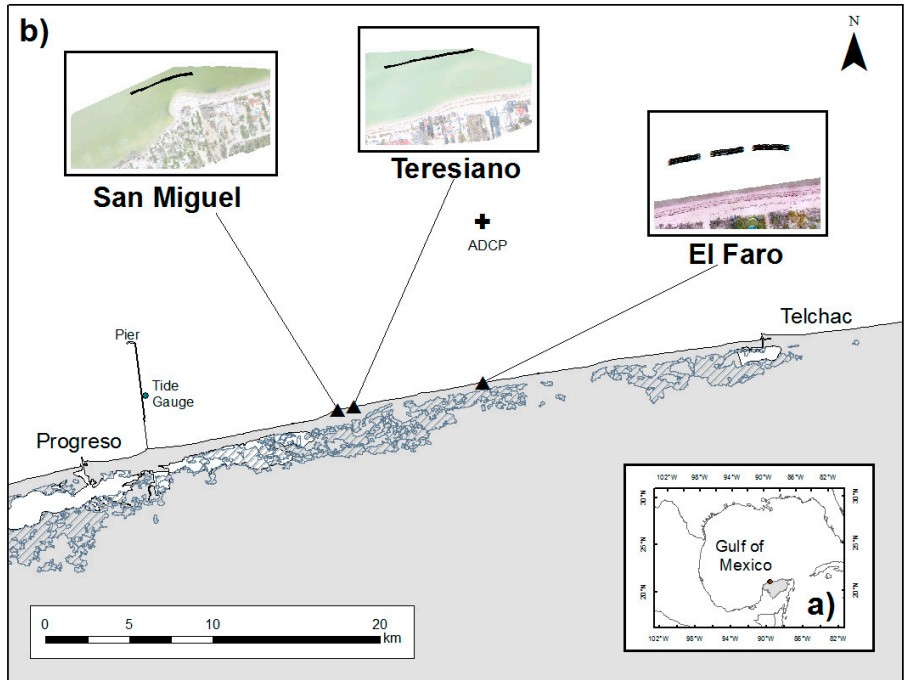

**Figure 1.** Study area location: (**a**) the breakwaters are located in the northern Yucatán Peninsula in the Gulf of Mexico; (**b**) the three sites (San Miguel, Teresiano, and El Faro) are between the ports of Progreso and Telchac. Waves and tides were measured with an Acoustic Doppler Current Profiler (ADCP) at a water depth of 10 m and a tidal gauge (www.mareagrafico.mx) installed near the 6 km-long Progreso Pier.

Low-crested structures were deployed between spring and summer 2017 in the northern Yucatán coast in Punta San Miguel, El Teresiano, and El Faro, located 10.5 km, 12 km, and 20 km from the Port of Progreso, respectively (Figure 1). The structures are all located within 9.5 km of the coast, and hence, environmental conditions (wind, waves, and water levels) are homogenous at all sites. Furthermore, the beach is characterized by the presence of beach houses constructed close to the shore (<15 m) over the primary dune.

San Miguel is located in a natural rocky headland where the shoreline orientation changes from 54 to 85° north (Figure 1). A 60-m rock revetment extends westward from the headland as a coastal protection measure to mitigate beach erosion. El Teresiano beach is located 1.5 km eastward from San Miguel and presents a uniform shoreline orientation of 80°. The beach width is less than 5 m at the most critical location, and waves reach the beach houses during high tides. Impermeable detached submerged breakwaters, made of geotextile tubes filled of sand, were installed at these two sites to increase the beach width (Figure 1).

El Faro has a shoreline orientation of 84° north (Figure 1), and the beach width at this site is <10 m. The coast is relatively pristine at the up-drift and down-drift side of the structure, but two beach houses are located behind the structure. The LCDBs deployed in El Faro are permeable.

*2.2. Breakwater Characteristics*

The low-crested detached breakwaters evaluated here are classified as impermeable and permeable based on the materials used for their construction. Differences in nearshore wave

transformation and sediment transport patterns were expected on the lee side of the structures owing to the distance to the undisturbed shoreline, freeboard height, breakwater length and orientation, and shoreline orientation (Figure 2). The impermeable breakwaters are made of sand-filled geosystems, and the transmissivity depends only on the wave propagation above the freeboard crest. On the other hand, the artificial reef modules known as Reef Balls™ [30] are considered to be permeable structures, because they allow flow through the modules and the two 10-m gaps separating the three breakwater segments. An overview of the breakwaters' characteristics for each site is given in Table 1 and described in more detail below.

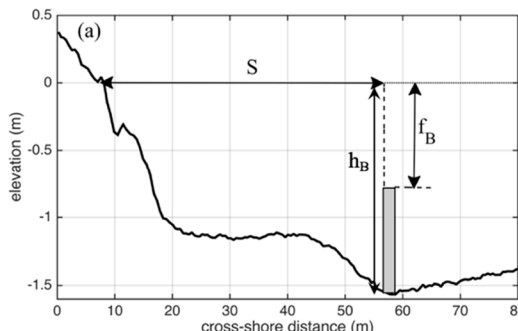
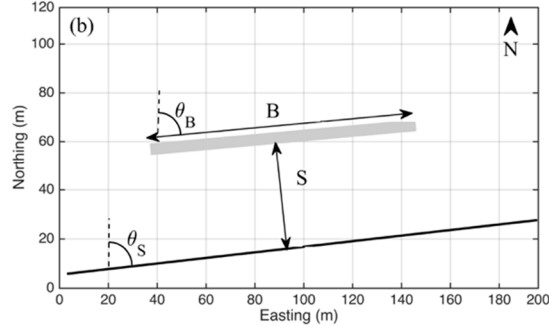

**Figure 2.** Description of structural parameters: (**a**) distance to the undisturbed shoreline (S), water depth at the breakwater ($h_B$), and freeboard elevation ($f_B$); (**b**) breakwater length (B), breakwater orientation ($\theta_B$), and undisturbed shoreline orientation ($\theta_S$).

**Table 1.** Breakwater characteristics at the three sites: breakwater length (B), distance from the shoreline (S), breakwater orientation ($\theta_B$), shoreline orientation ($\theta_S$), freeboard ($f_B$) of each section, and water depth at the breakwater ($h_B$).

| Location | B (m) | S (m) | $\theta_B$ (°) | $\theta_S$ (°) | $f_B$ (m) | $h_B$ (m) |
|---|---|---|---|---|---|---|
| San Miguel | 120 | 60 | 70 | 54 and 84 | −0.58, −0.55, −0.64, −0.39, −0.37, −0.36 | 1.3 |
| El Teresiano | 140 | 90 | 80 | 80 | −0.68, −0.68, −0.94, −1.02, −0.96, −0.69, −0.48, −0.62 | 1.6 |
| El Faro | 107 | 48 | 82 | 84 | −0.89, −0.88, −0.78 | 1.5 |

### 2.2.1. Impermeable Breakwaters

The impermeable structures were installed at San Miguel and El Teresiano beaches (Figure 1). The structures were constructed with 10- and 20-m sections of propylene geotextile mesh filled with sand with a minimum resistance of $50 \times 105$ kN/m, 415 gr/m$^2$, and 1.3 mm thickness. According to the design, each section has a width of 1.83 m and a height of 0.90 m when filled to 70% of its capacity.

The breakwater at San Miguel was composed of 6 sections and had a total length of 120 m with a minimum distance of 60 m to the original shoreline position (Figure 3a). The structure was oriented 70° north. The three easternmost 20-m sections of the breakwater at San Miguel beach were constructed from 18 June to 25 June 2017. The fourth section was installed on 14 July 2017, and in the following days, the remaining two sections were deployed. The structure required 620 m$^3$ of sand, which was procured on site. The freeboard elevation (vertical distance between mean sea level and breakwater crest) in San Miguel varied between $z = -0.36$ and $z = -0.64$ m.

The LCDB at El Teresiano, composed of 8 sections, had a total length of 140 m with a distance to shore of 90 m and was oriented parallel to the shoreline (see Table 1). The structure required 720 m$^3$ of sand, which was taken from the submerged beach between the structure and the shoreline, at a water depth of 1.5 m. The structure construction took place from 8 May to 16 May 2017. The freeboard presented significant differences between each section of the breakwater. At the time of deployment, the breakwater at El Teresiano had a freeboard that varied between $z = -0.48$ and $z = -1.02$ m along the structure. This heterogeneity was not consistent with the design. Figure 3 shows the layout of the two impermeable structures.

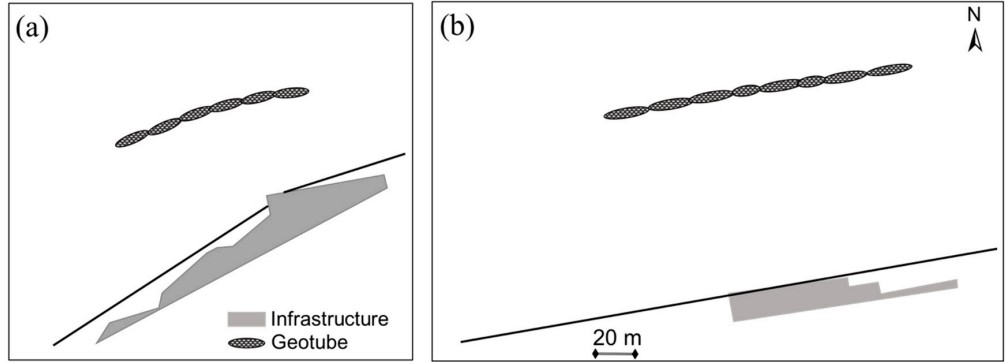

**Figure 3.** Impermeable detached breakwaters' layouts with respect to the mean shoreline orientation (black solid line) at (**a**) San Miguel and (**b**) El Teresiano.

### 2.2.2. Permeable Breakwaters

The permeable LCDB installed at El Faro beach was built using Reef Balls™. These are perforated hemispherical-shaped modules, available in different sizes, made of pH-neutralized concrete designed by Reef Ball Development Group, Ltd (Sarasota, FL, USA). The breakwater at El Faro has a total length of 107 m and consists of three 29-m sections separated by 10-m gaps (Figure 4). The structure was built with 135 elements of pallet balls arranged in two rows. The pallet balls are 800-kg elements, 0.90 m high, and with a 1.22-m diameter. The structure is oriented parallel to the shoreline, 48 m offshore (Figure 4). This field of detached breakwaters is considered, in this study, to be a single (permeable) breakwater.

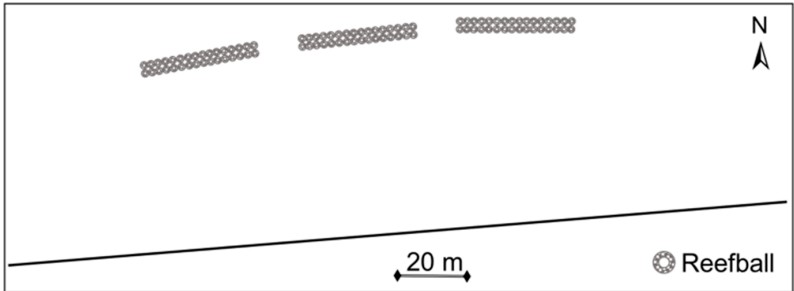

**Figure 4.** Reef Balls layout with respect to the mean shoreline orientation (black solid line) at El Faro.

## 3. Materials and Methods

Beach surveys were undertaken before the structure deployment in the three sites, and hence, beach morphology evolution could be evaluated with respect to the undisturbed condition. The freeboard elevation and breakwater length were periodically measured to assess structure variability. Furthermore, beach surveys and unmanned aerial vehicles (UAVs) flights were conducted to assess the functionality and far-field impact on adjacent beaches, respectively. Water levels and wave conditions were also measured to correlate the observed beach changes with the forcing conditions.

### 3.1. Beach Surveys

Beach surveys were conducted by means of a Leica™ real-time kinematics differential global positioning system (RTK-DGPS). Cross-shore transects were surveyed in the down-drift and up-drift areas and on the lee side of each structure with high spatial resolution. The GPS base and radio were installed inland on a fixed structure, specially built for this study, at each site, with known $x$, $y$, and $z$ coordinate locations. The rover DGPS was installed in a backpack to facilitate walking along each transect, starting from the dry beach and extending offshore until reaching a water depth of approximately 1.6 m. The beach surveys covered an alongshore stretch ranging from 300 to 400 m,

decreasing the separation between transects to 10 m on the lee side of the structures and increasing the separation to 20 m near the up-/down-drift boundaries of the surveyed area. A control point was measured at the beginning and at the end of each survey to correct the rover height on the backpack due to probable small vertical variations in its position during each survey. Typically, 20 cross-shore transects were surveyed at each site. However, the number of transects was increased in San Miguel from 20 to 27 due to the significant down-drift erosion observed beyond the original surveyed area (Figure 5).

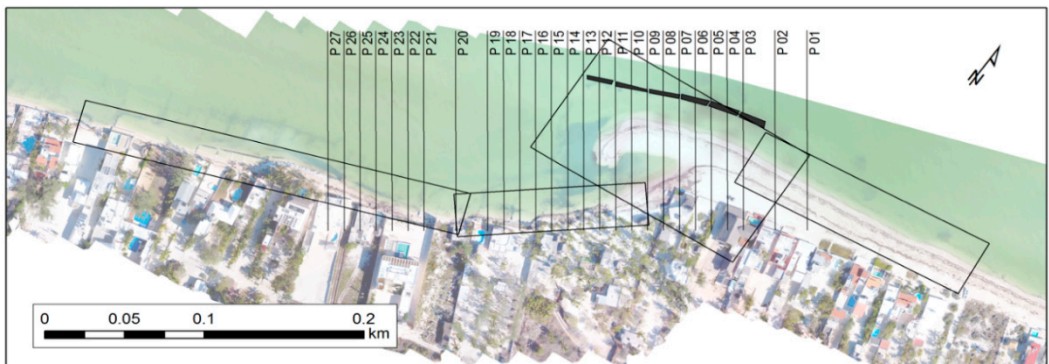

**Figure 5.** Aerial picture of San Miguel showing the survey transects (lines P01–P27) and the unmanned aerial vehicle (UAV) flight missions (rectangular shapes) employed for the monitoring. The aerial picture was taken on 11 April 2018.

The first survey at each site was taken as a reference to estimate relative sand volume changes. The shoreline position was estimated to be the cross-shore location with an elevation $z = 0$. Additionally, three-dimensional (3D) measurements allowed us to estimate the seabed changes in both the aerial and submerged beach profiles. A total of 43 beach surveys were conducted with approximately 14 surveys at each site (see symbols in Figure 6).

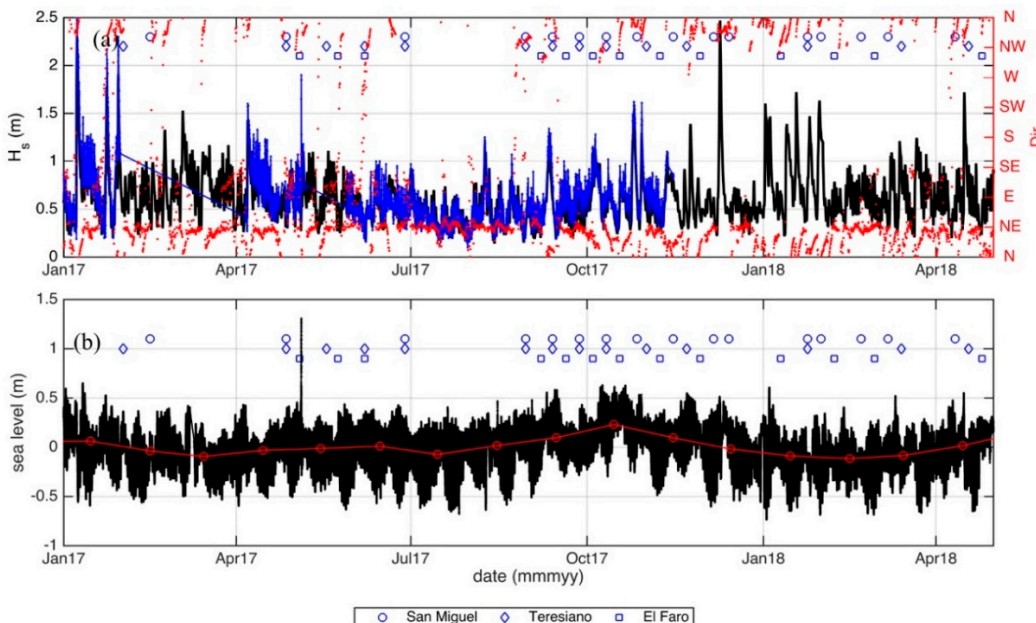

**Figure 6.** Time series of (**a**) significant wave height (black solid line: Wave Watch III hindcast; blue solid line: measured) and wave direction; and (**b**) mean sea level (black solid line: 1-min average; red circles: monthly average) during the study period. The blue symbols represent the surveys conducted at each site.

UAV flights have become a useful instrument in coastal studies [31]; the predominant advantages, including larger surveyed areas than GPS, graphic documentation through ortho-mosaics of the surveyed area, repeatability, and low cost, and, with the use of ground control points, centimeter-level precision digital surface models (DSMs) can be achieved. In this study, a Phantom 4 DJI quadcopter was used in combination with Pix4D Capture for flight planning and Pix4D Mapper to obtain digital surface models and subsequently extract the shoreline ($z = 0$).

Flights were conducted on a monthly basis for each site and when a scheduled engineering intervention or a storm event occurred. The mapped areas were rectangular with designed fixed parameters, such as a double grid to correctly compute the 3D models, 30-m flight altitude, 75% image overlap, 90-degree camera position, and medium-speed flight. Depending on the site and the complexity of the area, a series of missions was used (e.g., San Miguel was covered by 4 separate missions in each survey; see rectangles in Figure 5).

The accuracy of the ortho-mosaics and the DSMs depends largely on the quality and the coverage of the ground control points (GCPs) given that the surveyed images did not include a precise UAV location. Therefore, an average of 25 GCPs, precisely measured using RTK-DGPS, was used in each flight. Horizontal coordinates were referenced to the World Geodetic System of 1984 (WGS 1984) using the Universal Transverse Mercator coordinate system (UTM)Zone 16 N, and the vertical values were referenced to mean sea level using Gravimetric Mexican Geoid version 2010 (GGM10). The accuracy of the DSMs was validated using the RTK-DGPS data.

The obtained DSMs were integrated into the ArcGIS geographic information system, from which the coastline was obtained ($z = 0$) and put into the Digital Shoreline Analysis System [32], from which the shoreline change rate was obtained over the study period.

### 3.2. Waves and Water Level

Waves were measured with an RDI acoustic doppler velocimeter installed at 10-m water depth and located approximately 10 km offshore (see Figure 1). Wave measurements are depicted by the blue line in Figure 6a. Gaps in the record, due to instrument malfunction, were filled out using National Oceanic and Atmospheric Administration (NOAA) Wave Watch III hindcast data (black line in Figure 6a), corresponding with the closest node from the instrument's location. Maximum wave heights often occur during late fall and early spring and are associated with cold-front events, which drive NNW swells (Figure 6a). On the other hand, predominant, short-period NE waves associated with sea breeze events are present all year but are more frequent and intense during the spring–summer months, when cold fronts are absent (Figure 6a and see Figure 2 in [23]). Mean sea level was recorded with a tidal gauge from the National Mareographic Service (www.mareografico.unam.mx) located in the Progreso Pier (see Figure 1). A local storm called "turbonada" increased the mean sea level significantly on 4 May 2017 (Figure 6b).

## 4. Results

Environmental conditions, structure variability, and anthropogenic actions play an important role in beach morphodynamics. Thus, high spatial and temporal resolution monitoring allows us to understand a sea breeze-dominated beach's morphological responses to LCDBs. The analysis of the structure variability and the beach's morphological responses, based on field observations, is presented in this section.

### 4.1. Structures' Freeboard Variability

The impermeable LCDBs presented high spatial and temporal variability with respect to freeboard elevation. In the case of San Miguel, the deployment of the different sections was not uniform with the differences in elevation of 0.4 m between the sections (Figure 7a). Furthermore, the eastward sections' height changed up to −0.2 m in the two months following the installation due to loss of sand. The westmost section was torn apart in November 2017 to reduce down-drift erosion. By 11 April

2018, only the two middle sections remained, and by June, the breakwater was completely deflated (Figure 7a).

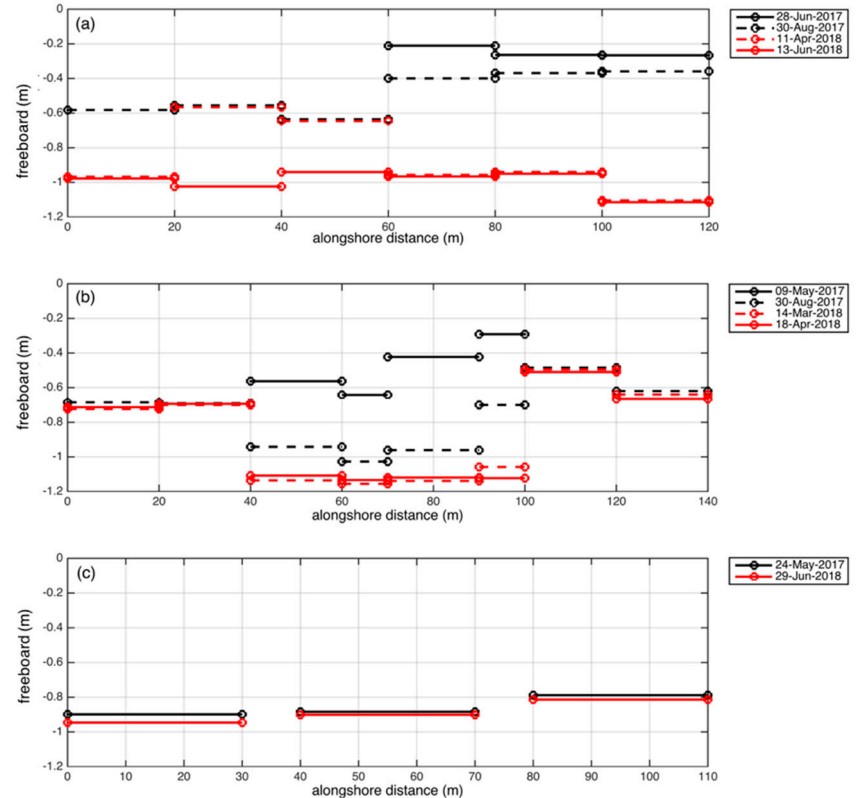

**Figure 7.** Freeboard elevation change at each of the sections of the low-crested breakwaters at (**a**) San Miguel, (**b**) El Teresiano, and (**c**) El Faro.

The breakwater at El Teresiano also presented significant differences with respect the freeboard height along the structure. The middle sections continuously lost elevation due to sand losses and were significantly reduced two months after deployment and almost completely deflated 10 months later. By 18 April 2018, only two 40-m sections remained with a gap between the sections of 60 m (Figure 7b). On the other hand, the breakwaters made of Reef Ball™ modules remined stable with freeboard changes of less than 0.10 m at the end of the study period (Figure 7c).

*4.2. Morphodynamic Response*

The importance of conducting high-resolution DGPS monitoring to explain beach changes is illustrated in San Miguel. The structure at this location was deployed with the aim of increasing the beach width at this natural headland to protect beach properties.

The beach morphology and structure history in San Miguel are shown in Figure 8. No significant changes were observed near the headland prior to the structure deployment (Figure 8a,b). The first half of the structure was deployed by 28 June 2017, and hence, some accumulation occurred at the eastward transects due to the net westward sediment transport (Figure 8c). The structure was fully deployed in July, and hence, the beach width increased behind the eastern half of the structure with low elevation, owing to the prevailing sea breeze conditions (Figure 8d). These conditions persisted until the beginning of the cold-front season, when swell waves decreased the salient length, increasing the beach elevation on the lee side of the structure and allowing sediment accretion in the western side (Figure 8e–i). However, significant beach erosion occurred at the down-drift (westward) beaches (Figure 8d–h). To mitigate down-drift beach erosion associated with sediment transport gradients due to wave diffraction, the westmost 20-m section of the geotextile was removed, inducing a positive

recovery effect (Figure 8i–k). However, in January 2018, the authorities decided to remove sand from the salient to nourish the down-drift beaches and prevent the formation of a tombolo. The estimated volume extracted was 600 m$^3$, which was taken from the subaerial beach (decreasing beach elevation) and was placed in the down-drift transects with low elevation (Figure 8l). The effect of this action was negligible one week later (Figure 8m), and the salient size increased during the following month (Figure 8n). During the following months, the eastward sections of the breakwater were torn apart (Figure 7a), gradually emptying and decreasing their freeboard height. Only two 20-m sections remained by April 2018. The resulting configuration of the LCDB induced the formation of a sand spit (Figure 8o), reducing the sediment supply and eroding the beach as far as 500 m in the down-drift areas. Table 2 summarizes the chronology of the main environmental and anthropogenic events controlling the morphological response at this site.

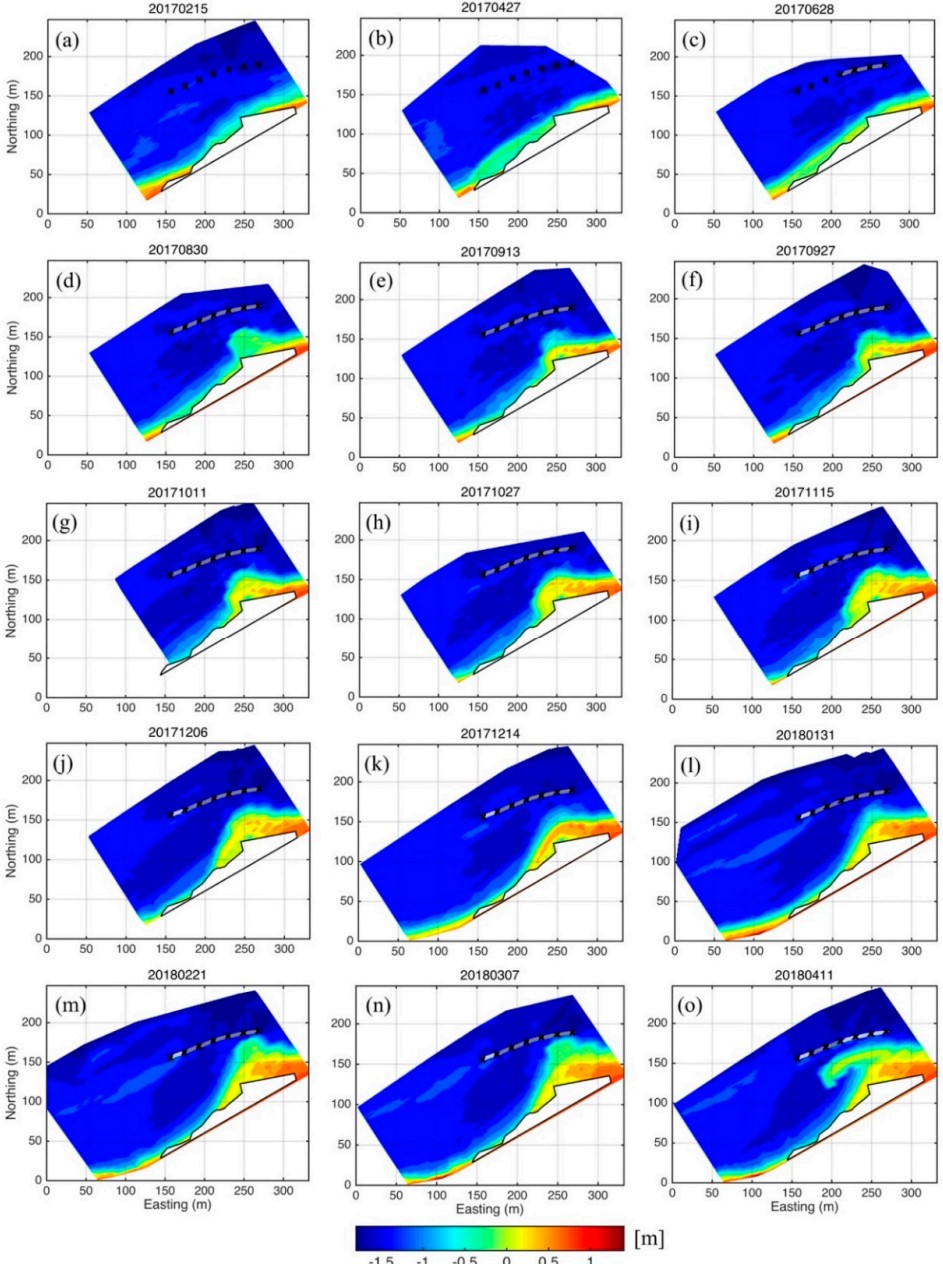

**Figure 8.** Topo-bathymetry and structure evolution at San Miguel from the (**a**) undisturbed condition (without structure) to (**o**) the last survey a year after. The title in each panel represents the survey date (*yyyymmdd*). The lighter color in breakwater sections indicates that the geotube section was deflated.

**Table 2.** Environmental and anthropogenic events affecting the beach evolution in San Miguel.

| Event | Dates | Morphology Effect |
|---|---|---|
| Low-Crested Detached Breakwater construction | June–July 2017 | Updrift accretion |
| Sea breeze events | July–August 2017 February–April 2018 | Salient growth in the eastern half |
| Central American Cold Surge events | September–December 2017 | Subaerial beach accretion and down-drift transport |
| Removal of westmost 20-m section | November 2017 | Recovery of down-drift beaches |
| Mechanical sand extraction from the salient and placement in down-drift area | January 2018 | No significant changes |
| Removal of eastward sections | March–April 2018 | Sand spit formation |

### 4.3. Spatial and Temporal Variability of Sand Volume

The volumetric changes were estimated using the high-resolution surveys from each site. The volume changes were computed for both the up-drift and down-drift areas taking as a reference the beach transect located at the center of the structure. Thus, the volume was integrated in the cross-shore (from the landward limit of each transect until h > −1.5 m) and alongshore direction (from the middle transect toward the down-drift and up-drift boundaries). The relative volume changes were estimated with respect to the reference volume obtained from the initial survey (without structures).

The up-drift (east, blue bars in Figure 9) area showed a volume increase in all sites, with a more significant increase with respect to the impermeable structures (Figure 9a,b). San Miguel beach showed a consistent accretion in the up-drift side, reaching more than 5000 m$^3$ in one year. Similar behavior was observed at El Teresiano but with a lower volume change (2500 m$^3$) over the same period. El Faro showed more sensitivity to seasonal changes, with a maximum accretion of 1500 m$^3$ but decreasing to less than 1000 m$^3$ by the end of the study period. On the other hand, the down-drift volume (red bars in Figure 9) showed a significant volume decrease with respect to the impermeable structures (Teresiano: 1000 m$^3$; San Miguel: 2000 m$^3$). The beach retreat in San Miguel was limited by a revetment that constrained the volume decrease. The permeable LCDB in El Faro, made of Reef Balls™, showed erosion at the down-drift side at about the same rate as the beach accretion during the first months. However, with the start of the cold-front season, the alongshore transport reversed, and the down-drift erosion decreased, turning into down-drift accretion at about the same volume as the up-drift accretion by the end of the study period (red bars in Figure 9c).

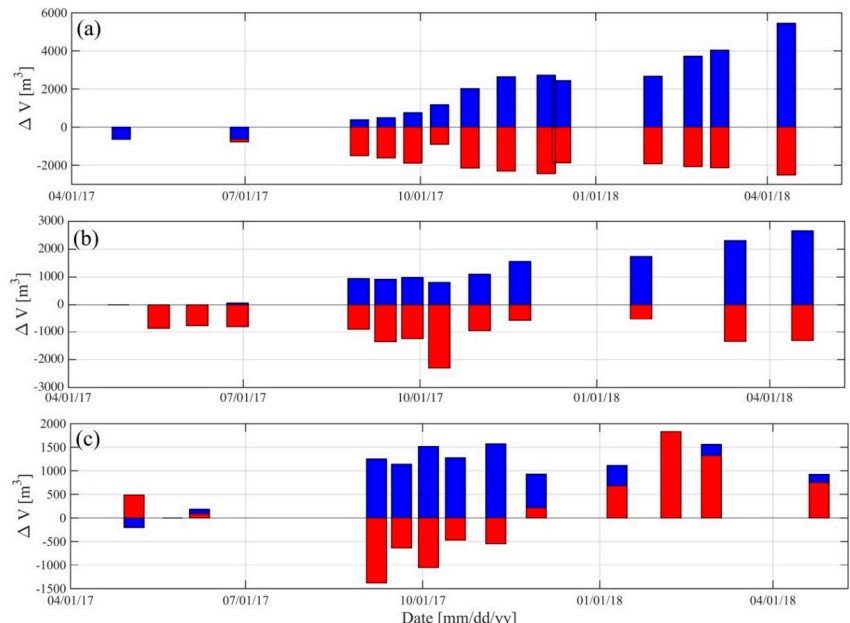

**Figure 9.** Relative volume change with respect to the LCDB central axis (east: blue; west: red) at (**a**) San Miguel, (**b**) El Teresiano, and (**c**) El Faro. Notice the difference in the vertical scales between panels.

The subaerial beach volume ($z > -0.5$ m) was also computed and evaluated at each beach profile (Figure 10). San Miguel showed a clear accretion/erosion trend in the eastern/western beach profiles (Figure 10a). This highlights the large impoundment produced by this LCDB. The negative impact on the down-drift side might have been enhanced by the drastic change in shoreline orientation at this site, large $B/S$ ratio, and high freeboard elevation. El Teresiano showed a moderate down-drift effect, which was attenuated by the end of the study period due to changes in the structure's elevation (Figure 10b). However, with the start of the sea breeze season, the accretion increased in the up-drift area. The LCDB made of Reef Balls™ modules showed a negative effect in the down-drift side, followed by a sustained volume increase at the beginning of the cold-front season. The accretion induced by the permeable structure was moderate as compared with that of the impermeable ones.

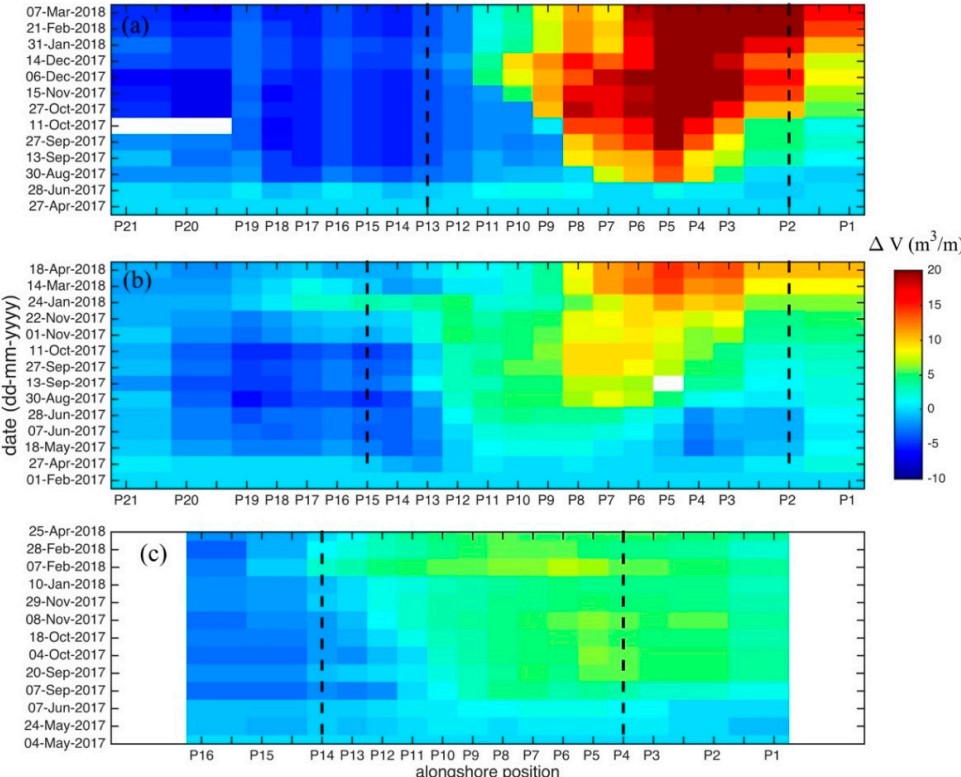

**Figure 10.** Subaerial volume change with respect to the reference beach survey (initial date, without LCDB) at (**a**) San Miguel, (**b**) El Teresiano, and (**c**) El Faro. The dashed vertical lines denote the limits of the LCDBs at each site. The date of the reference beach surveys corresponds to the initial date in the y-axis.

*4.4. Trends: Shoreline, Volume, and Seabed Elevation*

The trends of the shoreline position, emerged beach volume, and seabed elevation were estimated for the three sites during the study period. The rates of shoreline change were calculated using the shoreline ($z = 0$) extracted from the UAV flight data (San Miguel and El Teresiano) and DGPS surveys (El Faro). The rate of volume change and seabed elevation were calculated for the emerged section of each DGPS surveyed beach profile, considering $z = -0.5$ m to be the depth of the end of the emerged beach. The three beaches showed similar behavior with respect to trends in the shoreline position and in the emerged beach volume, with positive values in the eastern sections of the beach and less positive or negative values in the western sections. However, the magnitude and extension of the region with positive trends varied widely between sites.

San Miguel beach (Figure 11) showed the highest positive and negative shoreline trends of the three beaches. In the shadow area of the eastern section of the structure, the change rate was as high as 30 m/year. On the other hand, in the western section, the shoreline trends were negative—even in the

shadow area of the structure—with values as low as −20 m/year in the down-drift section (Figure 11b). Although the original flight path considered approximately 350 m on the western side of the structure, there was clear evidence that the negative trend extended as far as 500 m in the down-drift side. The change in the emerged beach volume showed similar trends with values ranging between +39.1 and −8.3 m³/m/year (Figure 11c). The beach presented a large accumulation of sediment in the eastern section of the study area. Figure 11d presents the rate of change in the elevation between 27 April 2017 (before the deployment of the structure) and 7 March 2018. In certain regions, the increase in beach elevation over the course of one year was larger than +1.5 m near the salient, whereas the decrease in the down-drift (western) locations reached values larger than −1.0 m.

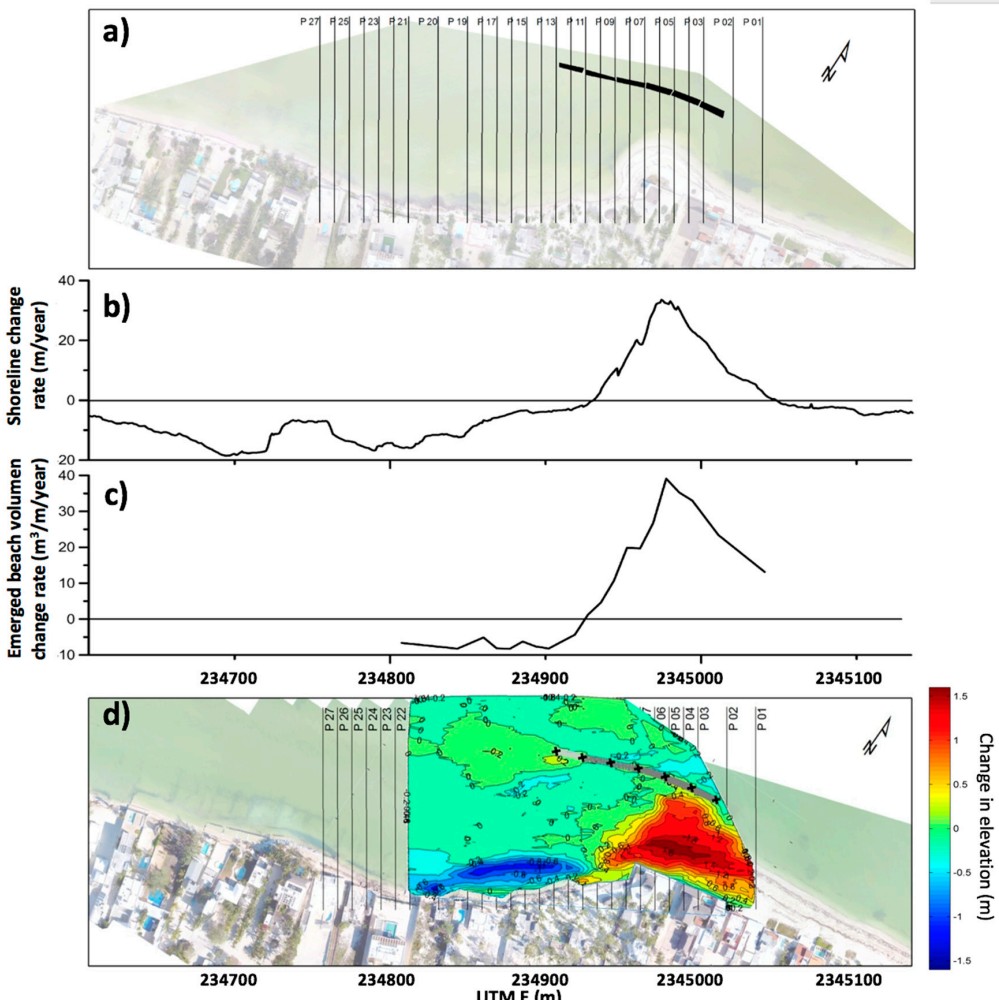

**Figure 11.** San Miguel beach (**a**) mosaic of the section of beach covered by the UAV flights with the location of the impermeable structure highlighted in black and parallel lines marking the high-resolution differential global positioning system (DGPS) profiles; (**b**) rate of change of the shoreline position during the study period obtained from UAV surveys; (**c**) rate of change of the emerged beach volume during the study period obtained from DGPS surveys; and (**d**) change in elevation between the last (7 March 2018) and first (27 April 2017, no structure) DGPS surveys.

El Teresiano (Figure 12) displayed a similar pattern, although (i) positive trends showing an advance in the shoreline position were below 20 m/year, (ii) the region where the shoreline trends were positive covered a wider section along the beach, and (iii) the negative trends also showed smaller values (>−5 m/year, Figure 12b). Regarding the volume, the trends varied between +10.8 and −3.5 m³/m/year (Figure 12c). As presented in Figure 12d, the increase in elevation between 27 April

and 18 April 2018 reached values similar to that in San Miguel (Figure 11d). However, the maximum decrease in elevation near the shoreline and in the emerged beach was found to be −0.4 m, less than half that in San Miguel.

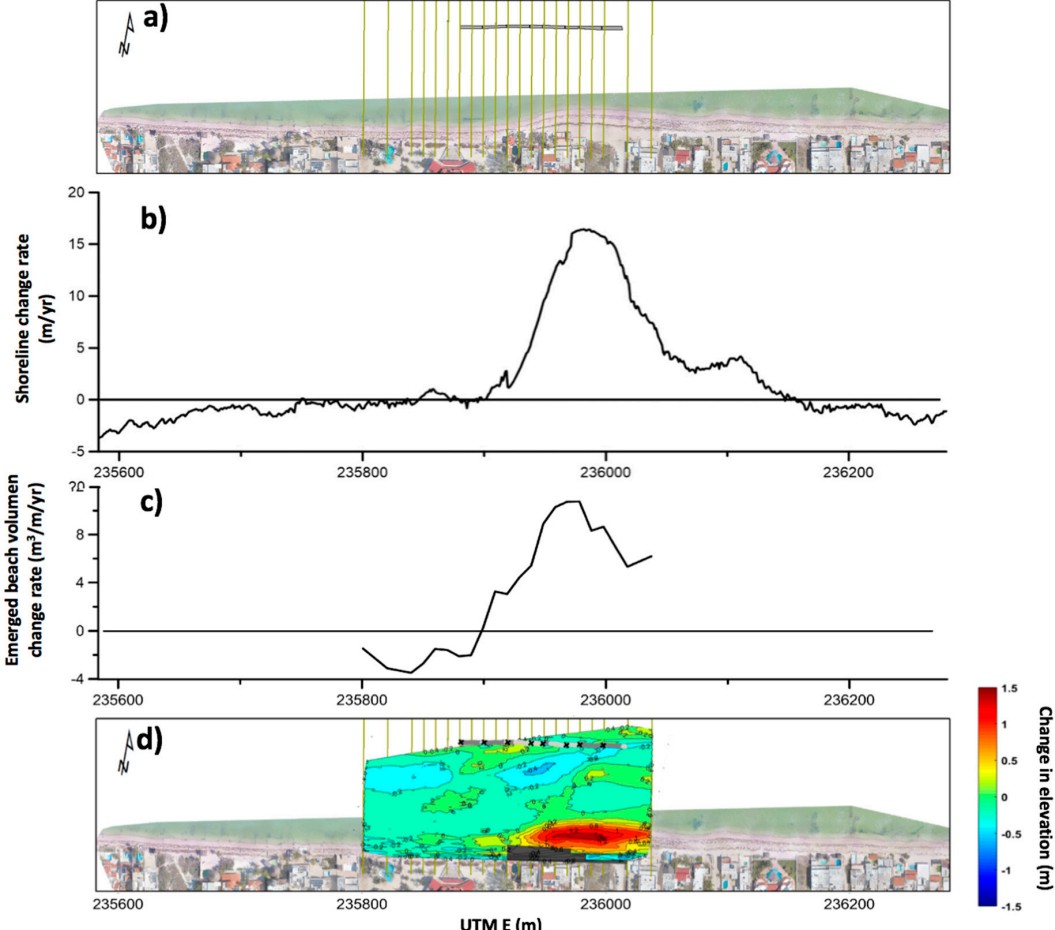

**Figure 12.** El Teresiano beach (**a**) mosaic of the section of beach covered by the UAV flights with the location of the impermeable structure highlighted in gray and parallel lines marking the high-resolution DGPS profiles; (**b**) rate of change of the shoreline position during the study period obtained from UAV surveys; (**c**) rate of change of the emerged beach volume during the study period obtained from DGPS surveys; and (**d**) change in elevation between the last (18 April 2018) and first (27 April 2017, no structure) DGPS surveys.

As for El Faro (Figure 13), the shoreline advance trends were positive on the lee side of the structure, with values below 6 m/year (Figure 13b). The trends in the emerged beach volume between 4 May 2017 and 25 April 2018 varied between +8.0 and −4.3 m$^3$/m/year. Changes in elevation after the deployment of the structure were less important at this site, with maximum increases of +0.6 m and decreases of −0.2 m.

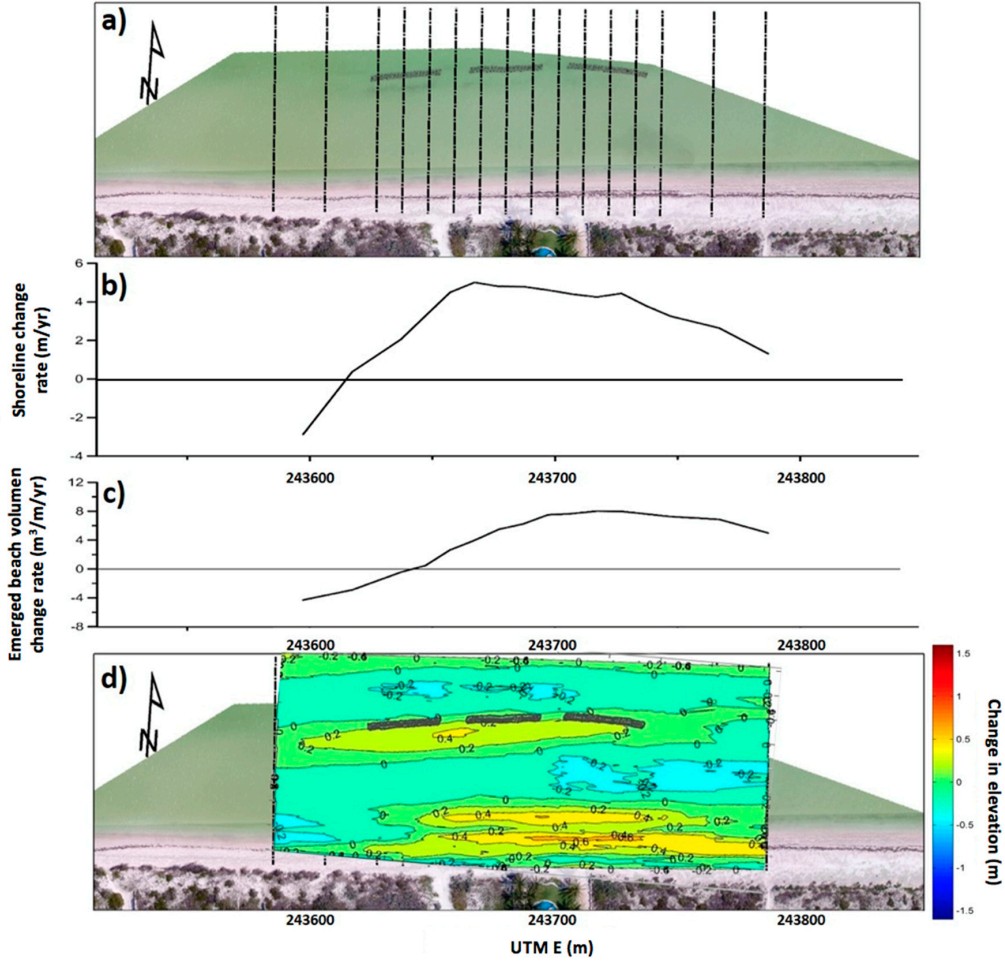

**Figure 13.** El Faro beach (**a**) aerial picture taken by the UAV flights with the location of the permeable structure highlighted in gray and parallel lines marking the high-resolution DGPS profiles; (**b**) rate of change of the shoreline position during the study period obtained from DGPS surveys; (**c**) rate of change of the emerged beach volume during the study period obtained from DGPS surveys; and (**d**) change in elevation between the last (25 April 2018) and the first survey (4 May 2017, no structure) DGPS surveys.

## 5. Discussion

A major problem with LCDB design is the difficulty of predicting the morphodynamic response on the lee side of the structure [7]. Empirical formulations, relating the distance to the tip of the salient $X_{off}$, the length of the structure $B$, and the distance to the undisturbed shoreline position $S$, predict a power curve relationship given by the following:

$$X_{off} = aB \left( \frac{B}{S} \right)^{b}$$

where the size of the salient $Y_s = S - X_{off}$ and the parameters $a$ and $b$ are those proposed by [18] and [17] for a single emergent breakwater ($a = 0.68$ and $b = -1.22$), reefs ($a = 0.50$ and $b = -1.27$), and islands ($a = 0.40$ and $b = -1.52$), respectively.

A distinct morphological response of the beach salient was observed in the three sites. The maximum shoreline salient size was measured in San Miguel, followed by El Teresiano and El Faro, respectively (Table 3 and Figures 11b, 12b and 13b). Empirical formulations (e.g., [17,18]) were employed, finding a satisfactory agreement for San Miguel (Table 3). This suggests that the beach response associated with a sand-filled geosystem can be predicted by the formulation developed by [17] for reefs. However, large differences between the observations and model predictions for El

Teresiano were found (Table 3). The latter can be ascribed to the continuous loss of sand in the middle sections of the geosystem (Figure 7b), which transformed a 140-m long breakwater into two 40-m long breakwaters separated by a 60 m gap. Applying the model developed by [17] to the case of El Teresiano for a 40-m breakwater predicted the size of the salient accurately (i.e., $Y_s$ = 20 m). On the other hand, the shoreline salient on the lee side of the breakwater at El Faro was not predicted by any model, owing to the high transmissivity through the modules and the gaps between the sections. The formulations by [18] consistently underpredicted the salient size for impermeable LCDBs.

**Table 3.** Measurements and predictions of shoreline salient size $Y_s$ at the three sites.

| Location | Ys (m) | Hsu and Evans (1990) | Islands in Black and Andrews (2001) | Reefs in Black and Andrews (2001) |
|---|---|---|---|---|
| San Miguel | 33 | 25 | 43 | 35 |
| El Teresiano | 16 | 34 | 61 | 50 |
| El Faro | 5 | 20 | 35 | 28 |

Sand-filled geosystems are highly vulnerable to vandalism in this area, and hence, the useful life of the structure can be drastically reduced. Furthermore, the tearing apart of the geotube sections plays an important role in beach evolution, leaving behind emptied geotextiles that are difficult to remove; moreover, the degradation of such geotextiles might have a negative ecological impact. It is not clear how the effect LCDBs with sand-filled geosystems can be correctly predicted if they are so prone to tearing and subsequent deflation.

## 6. Conclusions

A micro-tidal, sea-breeze-dominated beach's morphological responses to both impermeable and permeable low-crested detached breakwaters was analyzed based on high-resolution field observations. The following conclusions were found:

(1). Impermeable structures induced a salient located on the up-drift (east) side behind the structure, as well as down-drift erosion. The position and size of the salient is highly sensitive to the breakwater orientation with respect to the shoreline. Predicting the performance of sand-filled geosystems LCDBs is challenging owing to the high spatial and temporal variability of the structure.

(2). The permeable LCDB induced an order of magnitude less accretion and erosion than the impermeable structures, with a more symmetric salient on the lee side of the structure. The structures were found to remain stable during the study period. Based on observations, permeable (high transmissivity) structures are more suitable for the study area due to the persistent alongshore transport.

(3). The subaerial beach volume increase on the lee side of the LCDBs was strongly correlated with the beginning of the Central America cold surge season (i.e., October), owing to the combination of the maximum mean sea level and swell conditions. On the other hand, the freeboard elevation changes in the geotube sections showed no correlation with high energy conditions and hence can be ascribed to external factors.

(4). High spatial and temporal resolution measurements, combining DGPS and UAVs flights, were found to be important to explain far-field morphological changes.

(5). The design of sand-filled geosystems can be approximated with the formulation of [17] developed for reefs. On the other hand, the development of new formulations for high transmissivity structures, such as LCDB made of Reef Balls™ modules, is warranted.

**Author Contributions:** Conceptualization, A.T.-F., G.M., E.T.M., and E.O.; Original Draft Preparation, A.T.-F., G.M., E.T.M., and E.O.; Review and Editing of Manuscript, A.T.-F., G.M., E.T.M., E.O., and P.S.; Field Work, A.T.-F., G.M., E.T.M., and E.O.; Field Data Analysis, G.M., E.T.M., E.O., and A.T.-F.; and Funding Acquisition, P.S. and A.T.-F.

**Funding:** This research was funded by the Yucatán State Environmental Ministry of Yucatán and the Laboratorio Nacional de Resiliencia Costera (Project LN 293354). Additional financial support was provided by PAPIIT DGAPA UNAM (IN101218) and Investigación Científica Básica CONACYT (CB-2016-01-284819).

**Acknowledgments:** We are thankful for the use of Pix4D mapper Edu. NOAA is acknowledged for making Wave Watch III hindcast data available. The authors would like to thank Gonzalo Martín for IT support and José López Gonzalez and Juan Alberto Gómez for field support. Furthermore, a special thanks to the homeowners of the houses in San Miguel, El Teresiano, and El Faro for allowing the installation of the GPS base station. Anonymous reviewers provided fruitful comments that improved the manuscript.

**Conflicts of Interest:** The authors declare no conflict of interest.

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
