# Peer review of "Morphodynamic Response to Low-Crested Detached Breakwaters on a Sea Breeze-Dominated Coast"

_water, doi:10.3390/w11040635_

Round 1
Reviewer 1 Report
Manuscript ID: water-466147
Title: On the performance and stability of low-crested detached breakwaters on a sea-breeze dominated coast
General comment:
The paper deals with the analysis of littoral drift and beach morphodynamics behind three different low-crested detached breakwaters. Two types of breakwaters have been analyzed, impermeable and permeable. As a positive aspect, the field data included are vast and accurate. The filed campaign has been intensive and the paper includes some interesting results. As main negative aspects, it is not either a real discussion about breakwater stability or performance, but a discussion about beach morphology changes due to the breakwater presence.
As a general comment, I don’t understand the inclusion of San Benito breakwater in the paper, since there are no analysis or results of this structure.
Specific comments:
Article title: In my opinion, the title is misleading since the paper deals with beach morphodynamics and littoral drift and there is just a small discussion about breakwater freeboard changes. When talking about “performance” and “stability”, breakwater researchers expected a discussion about changes in the wave field, stability analysis, etc. So I suggest the title must be changed.
Line 59: the state of the art is poor in terms of beach performance under LCDBs deployment. ¿There are similar experiences over the world? ¿How do these breakwaters behave?
Line 113: Including a cross-section drawing of the breakwaters, indicating the parameters B, S, theta_B, theta_S and f_B, would be helpful. In any case, in breakwater design, it is common to assign the letter “B” to the crest width and “Rc” to the freeboard (see Coastal Engineering Manual as an example).
Line 127: Misprint, “This heterogeneity was “nos”…”, must be corrected.
Line 146: The same as in comment 3, cross sections drawing or a sketch of the permeable breakwaters would be helpful.
Line 224: Including the wave direction in Figure 5a makes it unclear. A wave rose would be more clarifier.
Line 234: I don’t agree that “measuring the elevations of different sections along the breakwater” is a stability analysis. I suggest a change in the terms included in the paper, maybe “lost of sand inside the geotextile” would be more precise.
Line 250: I suggest a change in the name of the ordinate axis in Figure 6, from “elevation” to “freeboard”.
Line 252: Misprint, there is an extra point at the end of the caption.
Line 361: Misprint, the words “salient length” are together.
Line 387: Following my previous comments, I don’t agree that the paper includes a stability analysis of the breakwater. Please consider changing this sentence.
Author Response
WATER
Reply to the Reviewer 1
“On the performance and stability of low-crested detached breakwaters on a sea-breeze dominated coast” (Manuscript ID: water-466147)
by Alec Torres-Freyermuth, Gabriela Medellín, Ernesto Tonatiuh Mendoza, Elena Ojeda, & Paulo Salles
General comment:
The paper deals with the analysis of littoral drift and beach morphodynamics behind three different low-crested detached breakwaters. Two types of breakwaters have been analyzed, impermeable and permeable. As a positive aspect, the field data included are vast and accurate. The filed campaign has been intensive and the paper includes some interesting results. As main negative aspects, it is not either a real discussion about breakwater stability or performance, but a discussion about beach morphology changes due to the breakwater presence.
RESPONSE: We acknowledge Reviewer 1 for having given fruitful comments which helped us to improve the manuscript. All comments provided by the reviewer have been incorporated in the revised version of the manuscript. The revised manuscript is now focused on the morphodynamic response. Furthermore, a point-by-point response to all the reviewer comments is given below.
As a general comment, I don’t understand the inclusion of San Benito breakwater in the paper, since there are no analysis or results of this structure.
RESPONSE: The analysis of San Benito data was not included in the manuscript due to the lack of information about the undisturbed beach condition. Following the reviewer’s suggestion, San Benito is no longer included in the paper and Figure 1, abstract and methods have been revised accordingly.
Specific comments:
Article title: In my opinion, the title is misleading since the paper deals with beach morphodynamics and littoral drift and there is just a small discussion about breakwater freeboard changes. When talking about “performance” and “stability”, breakwater researchers expected a discussion about changes in the wave field, stability analysis, etc. So I suggest the title must be changed.
RESPONSE: We agree with the reviewer suggestion and hence the title in the revised manuscript was changed to:
“Morphodynamic response to Low-Crested Detached Breakwaters on a sea-breeze dominated coast”
Line 59: the state of the art is poor in terms of beach performance under LCDBs deployment. ¿There are similar experiences over the world? ¿How do these breakwaters behave?
RESPONSE: The literature review has been significantly improved in the revised version of the manuscript. Previous efforts regarding the study of stability, shoreline response, and ecological impact of low-crested structures have included in the Introduction. A review of monitoring and modelling studies are also presented (Lines 44-110). Furthermore, an analysis of the breakwaters’ performance with respect to parametric models is presented in the Discussion section (Lines 530-560).
Line 113: Including a cross-section drawing of the breakwaters, indicating the parameters B, S, theta_B, theta_S and f_B, would be helpful. In any case, in breakwater design, it is common to assign the letter “B” to the crest width and “Rc” to the freeboard (see Coastal Engineering Manual as an example).
RESPONSE: A plan view and cross-section drawing, denoting the parameters and their nomenclature, are now included in Figure 2 of the revised manuscript. We adopted the nomenclature proposed by Hsu and Silvester (1990).
Line 127: Misprint, “This heterogeneity was “nos”…”, must be corrected.
RESPONSE: The misprint has been corrected by “This heterogeneity is not consistent with” (see Line 221).
Line 146: The same as in comment 3, cross sections drawing or a sketch of the permeable breakwaters would be helpful.
RESPONSE: Figure 2 in the revised ms is applicable to both types of structures.
Line 224: Including the wave direction in Figure 5a makes it unclear. A wave rose would be more clarifier.
RESPONSE: We agree that a wave rose would be clearer in general terms. However, the wave direction was included in Figure 5a to show its seasonal variability during the study period. In the revised manuscript we refer the readers (Line 320) to a recent paper where the wave rose for the study area is shown (i.e., Figure 2 in Medellín et al., 2018).
Line 234: I don’t agree that “measuring the elevations of different sections along the breakwater” is a stability analysis. I suggest a change in the terms included in the paper, maybe “lost of sand inside the geotextile” would be more precise.
RESPONSE: The stability analysis term was removed from the manuscript and was replaced by structures freeboard variability. Also, the term “loss of sand inside the geotextile” was employed to describe the causes of such changes (Lines 30, 343 and 546).
Line 250: I suggest a change in the name of the ordinate axis in Figure 6, from “elevation” to “freeboard”.
RESPONSE: The name of the ordinate axis in Figure 6 (Figure 7 in the revised ms) has been changed accordingly.
Line 252: Misprint, there is an extra point at the end of the caption.
RESPONSE: The misprint has been corrected.
Line 361: Misprint, the words “salient length” are together.
RESPONSE: The misprint has been corrected (Line 396).
Line 387: Following my previous comments, I don’t agree that the paper includes a stability analysis of the breakwater. Please consider changing this sentence.
RESPONSE: The sentence has been corrected (see Lines 334, 339).

Reviewer 2 Report
This paper presents a study about two different types of low-crested detached breakwaters (LCDB), considering geotextile and concrete reef ball blocks, located in four sites along a coastal stretch, in México. The work is interesting, is in the scope of the journal and is well written. However, the goals of the work should be clearly stated at the introduction and a discussion section is missing, comparing the LCDB performance with other similar studies. Section 5, named “Discussion”, corresponds in fact to additional results.
Please, also consider the following specific comments:
· Avoid the general use of the first person: remove or replace words such “we” or “us”.
· Line 13: references to examples of the widely employed LCDB should be presented.
· The keywords should not be repeated in the title and thus, it is suggested to delete “low-crested detached breakwaters” and “sea breezes”, keeping only 5 keywords.
· Line 41: the 1986 reference is quite old to refer that hard engineering solutions are still popular practice.
· Lines 64 to 66 refer to the goals of the work? The objectives should be better specified in the introduction section. The aim of the study is repeated later, in lines 157 to 159!
· Section 2 should also include references to the longshore sediment transport volumes and past coastal erosion rates at the study area. What was the goal when the LCDB were built?... was it for coastal protection against erosion problems?
· Line 98: the distance between El Faro and San Benito is 4 km, but should be consistent with the sites locations (km +20 and km +23). According to the figure 1, both locations present a field of detached breakwaters (never referred as field of detached breakwaters), based on the gap between sections. Is this the reason to be considered permeable breakwaters, or are the blocks (reef balls) the reason for that? Please, clarify.
· Figure 1: what is the line representing near the tide gauge location? Is that the pear? How long is it?
· Table 1: following a geographic order, the line related to San Miguel should be before the one related to El Teresiano. Why is El Teresiano with 8 different fB values, if the LCDB just presents 6 sections? It seems that are some confusion between San Miguel and El Teresiano description (lines 121 to 135), when comparing with the values in Table 1 and Figure 2 schemes. Please check. To be consistent, the B value for El Faro should be 107 meters and San Benito should be 70 meters. What is the hB value?... is it the bottom level?... if negative, cannot be the water column height or depth! The same in line 260.
· Line 127: z=-1.00 should be z=-1.02, to be in accordance with the Table 1. The same for line 134 values.
· Lines 127 and 134: references to Figure 6 should be avoided in this part of the text.
· Line 187: Do UAV flight missions correspond to the rectangular shapes in the Figure 4?... this should be referred in the Figure 4 caption.
· Line 214: where is red line, I believe should be blue line.
· Line 243 (and 246): what is considered "completely destroyed"? Where the geotextile material is and what is the motive for the registered destruction?
· Lines 260-262: something is missing in the sentence. What are the two areas referred and how were the volumes integrated?
· Line 270: if a revetment is limiting the monitored behavior, that revetment should be described in the study area characterization.
· Lines 273-274. The downdrift accretion is a strange behavior that should be explained. What was the reason for that? Is there any relationship with the wave climate?
· Line 278: this paragraph represents the beginning of a spatial variability section (that should be named 4.2.2) or section 4.2.1 should be renamed as spatiotemporal variability.
· Figures 9 to 11: do the emerged beach volumes variation represents m3/year, or m3/year/m? Please clarify and check the text in accordance.
· Lines 329 and 336: permeable or impermeable?
· Line 335: refer to the study sites in the same way. El Teresiano, instead of Teresiano.
· In fact, section 5 is not a discussion section. It should be removed or presented as an additional type of results. No discussion is presented in the manuscript. No considerations about what happened to the geotextiles are presented.
· Conclusion 3) is referred to the alongshore transport but no considerations on that are presented.
· Conclusion 4) is not discussed in the manuscript.
Author Response
WATER
Reply to the Reviewer 2
“On the performance and stability of low-crested detached breakwaters on a sea-breeze dominated coast” (Manuscript ID: water-466147)
by Alec Torres-Freyermuth, Gabriela Medellín, Ernesto Tonatiuh Mendoza, Elena Ojeda, & Paulo Salles
This paper presents a study about two different types of low-crested detached breakwaters (LCDB), considering geotextile and concrete reef ball blocks, located in four sites along a coastal stretch, in México. The work is interesting, is in the scope of the journal and is well written. However, the goals of the work should be clearly stated at the introduction and a discussion section is missing, comparing the LCDB performance with other similar studies. Section 5, named “Discussion”, corresponds in fact to additional results.
RESPONSE: We acknowledge Reviewer 2 for his/her fruitful comments which helped us to improve the manuscript. The scope of the study is now more clearly stated (Lines 100-105). Furthermore, following the reviewer’s suggestion, the material from the discussion section was moved to the Results section (Lines 380-423) and the new Discussion section (Lines 530-560) is now focused on the comparison of observations with parametric relationships obtained by previous studies. The possible causes of the differences between observations and predictions are presented. A point-by-point response is given below.
Please, also consider the following specific comments:
· Avoid the general use of the first person: remove or replace words such “we” or “us”.
RESPONSE: The use of words “we” and “us” have been avoided in the revised version of the ms.
· Line 13: references to examples of the widely employed LCDB should be presented.
RESPONSE: The introduction was thoroughly revised to include more references regarding prior studies on LCDB (Lines 38-88).
· The keywords should not be repeated in the title and thus, it is suggested to delete “low-crested detached breakwaters” and “sea breezes”, keeping only 5 keywords.
RESPONSE: The keywords “low-crested detached breakwaters” and “sea breezes” have been removed from the keyword list.
· Line 41: the 1986 reference is quite old to refer that hard engineering solutions are still popular practice.
RESPONSE: The reference has been removed and the paragraph has been modified to focus the literature review on both conventional and non-conventional Low-Crested Structures.
· Lines 64 to 66 refer to the goals of the work? The objectives should be better specified in the introduction section. The aim of the study is repeated later, in lines 157 to 159!
RESPONSE: Following the reviewer suggestions, the objectives of the study are more clearly presented in the revised Introduction (Lines 101-105). Moreover, Lines 157 to 159 were removed to avoid redundancy.
· Section 2 should also include references to the longshore sediment transport volumes and past coastal erosion rates at the study area. What was the goal when the LCDB were built?... was it for coastal protection against erosion problems?
RESPONSE: Information regarding sediment transport volumes, erosion rates, and background on the construction of LCDBs is now included in Lines 113-155.
· Line 98: the distance between El Faro and San Benito is 4 km, but should be consistent with the sites locations (km +20 and km +23). According to the figure 1, both locations present a field of detached breakwaters (never referred as field of detached breakwaters), based on the gap between sections. Is this the reason to be considered permeable breakwaters, or are the blocks (reef balls) the reason for that? Please, clarify.
RESPONSE: We thank the reviewer for pointing this out. However, San Benito has been removed from the manuscript since the data was neither analyzed nor discussed. The reason for considering reef ball breakwater as a permeable structure is because it allows flow through both, the structure and the two 10-m gaps separating the breakwater segments. This aspect is clarified in Lines 182-185.
· Figure 1: what is the line representing near the tide gauge location? Is that the pear? How long is it?
RESPONSE: The line near the tide gauge location is the Progreso pier which is 6 km-long. This information is now included in the figure’s caption (Line 158).
· Table 1: following a geographic order, the line related to San Miguel should be before the one related to El Teresiano. Why is El Teresiano with 8 different fB values, if the LCDB just presents 6 sections? It seems that are some confusion between San Miguel and El Teresiano description (lines 121 to 135), when comparing with the values in Table 1 and Figure 2 schemes. Please check. To be consistent, the B value for El Faro should be 107 meters and San Benito should be 70 meters. What is the hB value?... is it the bottom level?... if negative, cannot be the water column height or depth! The same in line 260.
RESPONSE: The values in Table 1 have been revised, as well as in the manuscript to follow the geographic order. Now San Miguel is described before El Teresiano to be consistent and San Benito has been removed since is not discussed. On the other hand, the values have been corrected. San Miguel breakwater is composed of 6 sections whereas El Teresiano is composed of 8 sections. The freeboard values have been completed for San Miguel. Depth has been set as a positive value in Table 1.
· Line 127: z=-1.00 should be z=-1.02, to be in accordance with the Table 1. The same for line 134 values.
RESPONSE: The values have been changed accordingly.
· Lines 127 and 134: references to Figure 6 should be avoided in this part of the text.
RESPONSE: Done.
· Line 187: Do UAV flight missions correspond to the rectangular shapes in the Figure 4?... this should be referred in the Figure 4 caption.
RESPONSE: The information has been included in Figure 4 (Figure 5 in the revised ms) caption.
· Line 214: where is red line, I believe should be blue line.
RESPONSE: Red line has been changed for blue line (Line 313).
· Line 243 (and 246): what is considered "completely destroyed"? Where the geotextile material is and what is the motive for the registered destruction?
RESPONSE: The term “completely destroyed” was replaced by “completely deflated”, referring to the significant loss of sand owing to the tearing apart of the geotextile. We have included the possible causes and its implications in the Discussion section (Lines 555-560).
· Lines 260-262: something is missing in the sentence. What are the two areas referred and how were the volumes integrated?
RESPONSE: Additional information on how the volumes were integrated for both the up- and down- drift areas is included in the manuscript.
· Line 270: if a revetment is limiting the monitored behavior, that revetment should be described in the study area characterization.
RESPONSE: We agree. The description has been incorporated in the study area section (Lines 143-144).
· Lines 273-274. The downdrift accretion is a strange behavior that should be explained. What was the reason for that? Is there any relationship with the wave climate?
RESPONSE: The accretion is ascribed to the reversal in the longshore sediment transport during the CACS season. This information is now included in Lines 395-397.
· Line 278: this paragraph represents the beginning of a spatial variability section (that should be named 4.2.2) or section 4.2.1 should be renamed as spatiotemporal variability.
RESPONSE: The numbering of the sections has been revised.
· Figures 9 to 11: do the emerged beach volumes variation represents m3/year, or m3/year/m? Please clarify and check the text in accordance.
RESPONSE: We thank the reviewer for noticing this typo. Emerged beach volume is m3/year/m and hence the figures’ labels were revised.
· Lines 329 and 336: permeable or impermeable?
RESPONSE: Fixed. Permeable has been changed for impermeable.
· Line 335: refer to the study sites in the same way. El Teresiano, instead of Teresiano.
RESPONSE: Teresiano has been changed for El Teresiano.
· In fact, section 5 is not a discussion section. It should be removed or presented as an additional type of results. No discussion is presented in the manuscript. No considerations about what happened to the geotextiles are presented.
RESPONSE: The discussion section has been heavily revised to focus on the comparison of observations of the beach salient size with parametric relationships obtained in previous studies. The possible causes on the differences in the morphodynamic response with respect to parametric models is also discussed. The discussion section on the original ms was moved to the results section.
· Conclusion 3) is referred to the alongshore transport but no considerations on that are presented.
RESPONSE: The importance of alongshore transport in the study site is now highlighted in the description of the study area (Lines 117-119) to support conclusion 3.
· Conclusion 4) is not discussed in the manuscript.
RESPONSE: The conclusion 4 was re-written.
Reviewer 3 Report
The paper reports the results of a field campaign in the northern Yucatan Peninsula, aimed at investigating the effects on morphodynamics of two kinds of low-crested detached beakwaters, i.e. geotextile tubes and Reef Ball modules. The stability of the structures was also monitored.
The work is interesting, methods and results are clearly reported. I recommend this manuscript to be published, even if some revisions are needed.
In my opinion, the only weakness can be found in the introduction section. The paper touches a relevant issue, i.e. the soft engineering systems in coastal defense practice which are not properly introduced and discussed. Specifically, the functioning principles of both geotextile and reef ball and the advantages/disadvantages with respect to other soft and hard defence systems
(e.g. vegetation, drainage, low-crested detached rubble mound breakwater) have to be added, in order to provide readers with a sufficient background, by adding some relevant references (e.g. Engineering Modeling of Wave Transmission of Reef Balls, 2014; Shape optimization of geotextile tubes for sandy beach protection, 2008; Laboratory Investigation on the Evolution of a Sandy Beach Nourishment Protected by a Mixed Soft–Hard System, 2018). Moreover, nevertheless the literature cited in the work is relevant in the field, I suggest to add more recent bibliography on e.g. nourishments or design criteria for segmented breakwaters (citations [2-4]).
After adding the above-suggested literature review, Lines 48-61 could be moved in Section 2.
L. 66-71 refer to results rather than the introduction of the work. In the introduction, authors should describe the objectives of the work, by summarizing the methods and the phenomena (variables) investigated.
Other specific comments are listed below:
L.20 Performances are reported for 3 sites.
Keywords: I suggest to eliminate 'permeable and impermeable' and add 'beach morphodynamics'.
L.85-86: Distances in the brackets are not clear to what refers to.
L. 114: Table 1, 'distance from the structure to shoreline', please amend in 'distance from the shoreline'; specify the reference of the breakwater orientation; define before the table 1 the 'sections of the breakwaters'.
L.127-128: the different freeboard along the structure is due to the irregularities of bathymetry?
L.189: the title of the subsection 3.2 can be eliminated since it refers to beach surveys methods.
L. 212: field measurements were validated?
L. 214: The motivation of gaps in measurements has to be explained.
L. 238: please eliminate 'made of geotextile....with sand'. Impermeable structures have been yet defined in the previous section.
Figures 2 and 3 could be improved, for example by adding a section of the structure.
Author Response
WATER
Reply to the Reviewer 3
“On the performance and stability of low-crested detached breakwaters on a sea-breeze dominated coast” (Manuscript ID: water-466147)
by Alec Torres-Freyermuth, Gabriela Medellín, Ernesto Tonatiuh Mendoza, Elena Ojeda, & Paulo Salles
Comments and Suggestions for Authors
The paper reports the results of a field campaign in the northern Yucatan Peninsula, aimed at investigating the effects on morphodynamics of two kinds of low-crested detached beakwaters, i.e. geotextile tubes and Reef Ball modules. The stability of the structures was also monitored.
The work is interesting, methods and results are clearly reported. I recommend this manuscript to be published, even if some revisions are needed.
In my opinion, the only weakness can be found in the introduction section. The paper touches a relevant issue, i.e. the soft engineering systems in coastal defense practice which are not properly introduced and discussed. Specifically, the functioning principles of both geotextile and reef ball and the advantages/disadvantages with respect to other soft and hard defence systems (e.g. vegetation, drainage, low-crested detached rubble mound breakwater) have to be added, in order to provide readers with a sufficient background, by adding some relevant references (e.g. Engineering Modeling of Wave Transmission of Reef Balls, 2014; Shape optimization of geotextile tubes for sandy beach protection, 2008; Laboratory Investigation on the Evolution of a Sandy Beach Nourishment Protected by a Mixed Soft–Hard System, 2018). Moreover, nevertheless the literature cited in the work is relevant in the field, I suggest to add more recent bibliography on e.g. nourishments or design criteria for segmented breakwaters (citations [2-4]).
RESPONSE: We thank the reviewer for pointing out to the weakness of the first version of the manuscript. A thorough literature review has been included in the revised version of the manuscript. More specifically, updated references and relevant references suggested by the reviewer have been incorporated to present the functioning principles of the LCS and previous efforts devoted to improve the understanding of both conventional and non-conventional Low-Crested Structures. The new material is included in Lines 44-74.
After adding the above-suggested literature review, Lines 48-61 could be moved in Section 2.
RESPONSE: Following the reviewer’s suggestion, Lines 48-61 have been moved to Section 2 (Lines 120-134 in the revised ms).
L. 66-71 refer to results rather than the introduction of the work. In the introduction, authors should describe the objectives of the work, by summarizing the methods and the phenomena (variables) investigated.
RESPONSE: Following the reviewer’s comments, the text has been revised to present the objectives of this work more clearly (Lines 101-105).
Other specific comments are listed below:
L.20 Performances are reported for 3 sites.
RESPONSE: San Benito has been removed from the manuscript and hence Figure 1, the abstract, and description of the structures were revised accordingly.
Keywords: I suggest to eliminate 'permeable and impermeable' and add 'beach morphodynamics'.
RESPONSE: Keywords have been modified accordingly by adding “beach morphodynamics” and removing “permeable and impermeable”.
L.85-86: Distances in the brackets are not clear to what refers to.
RESPONSE: They refer to distances from Progreso. This has been clarified in the text (Lines 135-138).
L. 114: Table 1, 'distance from the structure to shoreline', please amend in 'distance from the shoreline'; specify the reference of the breakwater orientation; define before the table 1 the 'sections of the breakwaters'.
RESPONSE: The text has been revised in Table 1 and a new figure (i.e., Figure 2) has been included in the revised manuscript to clarify the reference of both the breakwater and the shoreline orientation.
L.127-128: the different freeboard along the structure is due to the irregularities of bathymetry?
RESPONSE: The structures should have been deployed with a constant freeboard according to the design, therefore, different freeboard along the structure is due to a deficient deployment when filling the geotextile tubes with sand.
L.189: the title of the subsection 3.2 can be eliminated since it refers to beach surveys methods.
RESPONSE: The title of the subsection 3.2 has been removed.
L. 212: field measurements were validated?
RESPONSE: The subaerial topography obtained using UAVs were validated with beach profiles acquired using RTK-GPS.
L. 214: The motivation of gaps in measurements has to be explained.
RESPONSE: We employed brand-new RDI ADCPs that presented a malfunction and stopped recording. This issue is now mentioned in the manuscript (Line 313).
L. 238: please eliminate 'made of geotextile....with sand'. Impermeable structures have been yet defined in the previous section.
RESPONSE: The text has been eliminated.
Figures 2 and 3 could be improved, for example by adding a section of the structure.
RESPONSE: We have included a new schematic (i.e., Figure 2) applicable to the two types of structures to clarify the parameters and nomenclature.
Reviewer 4 Report
Authors present the results of investigation of beach morphodynamics behind LCDBs deployed on a micro-tidal sea-breeze dominated beach (northern Yucatan peninsula). Numerical results are validated by comparison with experimental data obtained from a scaled physical model. High resolution Real Time Kinematics GPS beach surveys were conducted, along with Aerial Vehicle flights to evaluate the shoreline stability at adjacent beaches. According to observation results authors suggest that the study area is highly sensitive to the presence of impermeable LDCBs.
General comment
The objectives, methods, and major innovations of this study are clearly described.
It would be very benefitial to outline some conclusions on the influence of mid-sea level variations and higher energy wave episodes on the stability and functionality of LCDBs over the analyzed period.
Particular comments
Page 1, line 29
Please, replace “LDCBs” with “LCDBs”
Page 4, Table 1
Freeboard (fB) for San Benito +1.15 (please check). Presented data are related to the very beginig of field survay (fabruary 2017) or ?
Page 6, Figure 4.
Aerial picture of San Miguel indicate distance to the original shoreline position less then 60 m, as stated stated in the text (page 4, line 130). Please clerify.
Page 7, line 213-214
Wave measurements are depicted by the blue line (instead of „red“) in Figure 5a.
Page 10, Figure 8.
Please, write down the date of reference beach survey.
REVIEW CONCLUSION
The paper entitled “ON THE PERFORMANCE AND STABILITY OF LOW-CRESTED DETACHED BREAKWATERS ON A SEA-BREEZE DOMINATED COAST“ shows the results of the engineering correct methodological approach. The article will be suitable for publication in the journal “Water” after the corrections are made on current manuscript, according to the reviewer comments.
Author Response
WATER
Reply to the Reviewer 4
“On the performance and stability of low-crested detached breakwaters on a sea-breeze dominated coast” (Manuscript ID: water-466147)
by Alec Torres-Freyermuth, Gabriela Medellín, Ernesto Tonatiuh Mendoza, Elena Ojeda, & Paulo Salles
Comments and Suggestions for Authors
Authors present the results of investigation of beach morphodynamics behind LCDBs deployed on a micro-tidal sea-breeze dominated beach (northern Yucatan peninsula). Numerical results are validated by comparison with experimental data obtained from a scaled physical model. High resolution Real Time Kinematics GPS beach surveys were conducted, along with Aerial Vehicle flights to evaluate the shoreline stability at adjacent beaches. According to observation results authors suggest that the study area is highly sensitive to the presence of impermeable LDCBs.
General comment
The objectives, methods, and major innovations of this study are clearly described.
It would be very benefitial to outline some conclusions on the influence of mid-sea level variations and higher energy wave episodes on the stability and functionality of LCDBs over the analyzed period.
RESPONSE: Following the reviewer suggestion, we have included a conclusion regarding the influence of high energy and mid-sea level in both the performance and stability of the LCDBs:
“Subaerial beach volume increase in the leeside of the LCDBs was strongly correlated with the beginning of the CACS season (i.e., October) owing to the combination of maximum mean sea level and swell conditions. On the other hand, the freeboard elevation changes show no correlation with high energy conditions and hence can be ascribed to external factors.”
Particular comments
Page 1, line 29
Please, replace “LDCBs” with “LCDBs”
RESPONSE: The acronym has been corrected.
Page 4, Table 1
Freeboard (fB) for San Benito +1.15 (please check). Presented data are related to the very beginig of field survay (fabruary 2017) or ?
RESPONSE: The site of San Benito has been removed from the manuscript due to lack of information regarding the undisturbed beach condition to allow analysis of this site.
Page 6, Figure 4.
Aerial picture of San Miguel indicate distance to the original shoreline position less then 60 m, as stated stated in the text (page 4, line 130). Please clerify.
RESPONSE: The aerial picture corresponds to the beach conditions almost one year after the structure deployment (i.e., April 2018) and hence the shoreline position is less than the undisturbed shoreline located 60 m from the structure. The initial shoreline position is depicted in Figure 3a. To avoid confusion, the date of the flight has been included in the figure’s caption (Figure 5 in revised ms).
Page 7, line 213-214
Wave measurements are depicted by the blue line (instead of „red“) in Figure 5a.
RESPONSE: ‘Red line” has been changed for “blue line” to refer to the wave measurements in Figure 5a (Figure 6a in the revised ms).
Page 10, Figure 8.
Please, write down the date of reference beach survey.
RESPONSE: The date of the reference beach survey corresponds to the initial date indicated in the y-axis of each figure. This has been clarified in the Figure’s caption.
Round 2
Reviewer 2 Report
This paper presents a study about two different types of low-crested detached breakwaters (LCDB), considering geotextile and concrete reef ball blocks, located in three sites along a coastal stretch, in México. The work is interesting, is in the scope of the journal and is well written. Please, just consider the following specific comments:
· Line 93: the reference should be numbered [21], instead using the name of the author, to be in accordance with all the text in the manuscript.
· Line 100: check the distance between structures. 20km – 10.5km = 9.5km and not 12.5km!
· Figure 2: caption refers to hb, but this variable is not schematized in the figure.
· Lines 136: it is three sites instead of four!
· Table 1: B = 110m at El Faro should be 107m, to be consistent with the text, at line 165.
· Line 153: “the” instead of “th”.
· Line 230: Figure 6a instead of Figure 4a.
· Line 315: delete “beach” after “San Miguel”, to avoid the word repetition.
· Line 319: remove the space in “decreas ed”.
· Figure 11c: the vertical axis units should be m3/m/year.
· Line 413: check the number of the figure.
· Line 438: it is only one single permeable structure and thus, singular should be used.
· Line 442: refer the meaning of CACS!
Author Response
p.p1 {margin: 0.0px 0.0px 0.0px 0.0px; font: 12.0px Arial; min-height: 14.0px} p.p2 {margin: 0.0px 0.0px 0.0px 0.0px; font: 11.5px Arial; color: #0084cc} p.p3 {margin: 0.0px 0.0px 0.0px 0.0px; font: 11.0px Arial; color: #0084cc} p.p4 {margin: 0.0px 0.0px 0.0px 0.0px; font: 10.0px Arial} p.p5 {margin: 0.0px 0.0px 0.0px 0.0px; font: 10.0px Arial; color: #0084cc} span.s1 {font: 12.0px Arial; color: #000000} span.s2 {font: 11.5px Arial} span.s3 {font: 6.5px Arial}WATER
Reply to the Reviewer 2
“On the performance and stability of low-crested detached breakwaters on a sea-breeze dominated coast” (Manuscript ID: water-466147R2)
by Alec Torres-Freyermuth, Gabriela Medellín, Ernesto Tonatiuh Mendoza, Elena Ojeda, & Paulo Salles
This paper presents a study about two different types of low-crested detached breakwaters (LCDB), considering geotextile and concrete reef ball blocks, located in three sites along a coastal stretch, in México. The work is interesting, is in the scope of the journal and is well written. Please, just consider the following specific comments:
RESPONSE: We acknowledge Reviewer 2 for his/her thorough review of the manuscript. All comments have been incorporated in the revised version of the manuscript. A point-by-point response is provided below.
· Line 93: the reference should be numbered [21], instead using the name of the author, to be in accordance with all the text in the manuscript.
RESPONSE: The reference has been corrected following the journal guideline.
· Line 100: check the distance between structures. 20km – 10.5km = 9.5km and not 12.5km!
RESPONSE: The typo has been fixed and distance has been change to 9.5 km. The 12.5 km distance referred to the distance to San Benito site which has been removed from the manuscript.
· Figure 2: caption refers to hb, but this variable is not schematized in the figure.
RESPONSE: Figure 2 has been revised accordingly.
· Lines 136: it is three sites instead of four!
RESPONSE: Fixed.
· Table 1: B = 110m at El Faro should be 107m, to be consistent with the text, at line 165.
RESPONSE: Table 1 has been corrected.
· Line 153: “the” instead of “th”.
RESPONSE: Fixed.
· Line 230: Figure 6a instead of Figure 4a.
RESPONSE: Fixed.
· Line 315: delete “beach” after “San Miguel”, to avoid the word repetition.
RESPONSE: Done.
· Line 319: remove the space in “decreas ed”.
RESPONSE: Done.
· Figure 11c: the vertical axis units should be m3/m/year.
RESPONSE: The Figure 11 has been revised accordingly.
· Line 413: check the number of the figure.
RESPONSE: The number of the figure has been corrected.
· Line 438: it is only one single permeable structure and thus, singular should be used.
RESPONSE: The sentence has been revised.
· Line 442: refer the meaning of CACS
RESPONSE: The acronym CACS has been replaced by Central American Cold Surges.